# Coevolutionary dynamics of viruses and their defective interfering particles

Shiv Muthupandiyan, John Yin *

Wisconsin Institute for Discovery, Chemical and Biological Engineering, University of Wisconsin-Madison, Madison, Wisconsin, United States of America

* john.yin@wisc.edu

## Abstract

Defective interfering particles (DIPs) are viral mutants that arise naturally during infection. Because they lack one or more essential functions, DIPs cannot replicate on their own, but they can parasitize intact viruses during coinfection by competing for growth resources, thereby interfering with viral replication. The evolutionary interplay between viruses and their DIPs involves growth, mutation, interference, and resource trade-offs, but the mechanisms shaping population-level outcomes remain poorly understood. To address this, we developed a phenotype-space model across continuous traits (e.g., replicase binding affinity or packaging signal strength) using coupled partial differential equations that incorporate mutation, phenotype-dependent interference, intrinsic fitness costs, and *de novo* DIP generation. Unlike traditional strong-selection models, this framework captures strong-mutation regimes in which both virus and DIP populations evolve by diffusion through trait space and interact based on phenotypic similarity. Our analysis reveals two levels of dynamics. At the population level, viruses and DIPs undergo oscillations, consistent with predator–prey–like cycles (the von Magnus effect) observed experimentally. At the trait level, evolution drives shifts in resistance and interference, producing coevolutionary chases in which viruses temporarily escape and DIPs attempt to follow, as observed in serial-passage evolution studies. Systematic variation of parameters reveals four qualitative regimes: viral–DIP coexistence, sustained coevolutionary (Red Queen) chase dynamics, DIP extinction, and mutual extinction. Chase dynamics are most strongly promoted by intermediate interference strength and low decay rates, while higher levels drive collapse of one or both populations. The model further predicts thresholds where viral escape is either constrained by intrinsic fitness penalties or enabled through phenotypic separation from DIPs. These findings establish a general framework for virus–DIP coevolution, showing how both population dynamics and trait evolution shape outcomes, with implications for designing DIP-based therapeutics that better resist viral escape.

provided the original author and source are credited.

**Data availability statement:** No new data were generated in this study. All code used for simulations and analyses is publicly available at https://github.com/shivmuthupandiyan/virus-dip-coevolution.

**Funding:** This work was supported by funding from the US National Science Foundation (NSF) under grants DMS-2151959 (J.Y.), MCB-2029281 (J.Y.), and CBET-2030750 (J.Y.); the National Institutes of Health (NIH) under awards OT2OD030524 (J.Y.) and R01DK133605 (J.Y.). The funders had no role in study design, data collection and analysis, decision to publish, or preparation of the manuscript.

## Author summary

Most viruses produce defective versions of themselves, known as defective interfering particles (DIPs). DIPs cannot multiply on their own, but when they infect the same cell as an intact virus, they steal resources and limit the virus's growth. This makes them a promising antiviral therapy. But viruses may evolve to reduce the ability of DIPs to steal resources. We built a mathematical model that lets both viruses and DIPs vary in traits—such as how quickly they grow or how strongly they interfere—and evolve such traits over time. Our results reveal two layers of dynamic behavior. At the population level, viruses and DIPs can rise and fall in repeating cycles that resemble predator–prey interactions, even when traits stay fixed. With evolution, however, the contest gains another dimension: viruses may shift traits to escape interference, while new DIPs can arise and adapt to follow them. This coevolutionary chase is a novel feature that sets DIPs apart from conventional therapies, which cannot adjust to viral change. By exploring many conditions, we identified when viruses and DIPs coexist, when one eliminates the other, and when they remain locked in long-term pursuit. These insights suggest principles for designing therapeutic DIPs that might better resist viral escape.

## Introduction

Since the earliest days of virus cultivation, laboratory virus stocks contained aberrant particles that suppressed the replication of fully infectious virus. In a landmark series of experiments, Preben von Magnus showed that undiluted passage cultures of influenza virus led to the accumulation of non-infectious "incomplete" particles that interfered with standard virus replication [1–3]. This progressive enrichment of defective particles and corresponding drop in infectious virus output came to be known as the von Magnus effect [4]. Studies of vesicular stomatitis virus (VSV) and influenza virus showed how the von Magnus effect could drive predator–prey–like oscillatory dynamics between populations of defective interfering particles and standard virus [5–10].

Defective interfering particles, also called DI particles or DIPs, package and deliver defective virus genomes to host cells. The defective viral genomes are byproducts of error-prone genome replication by most RNA viruses, including VSV, influenza, and coronavirus, where their defects are most often linked to the deletion of one or more essential genes for virus growth [11–13]. Sequencing and analyses of environmental and clinical isolates has found defective genomes for many other RNA viruses, providing evidence for their occurrence and transmission in nature as well as their potential to modulate disease severity [14–18].

Early recognition of DIPs as natural parasites of virus replication spurred efforts to understand and exploit their interference for therapeutic benefit. In animal models, DIPs protected mice and ferrets challenged with RNA viruses such as VSV, Semliki Forest virus (SFV), and influenza A virus [19–21]. More recently, DIP-inspired

therapeutic interfering particles derived from influenza, polio, SARS-CoV-2, and HIV-1 have shown promise in rodents and non-human primates [22–25]. These therapeutic interfering particles (TIPs) may suppress the emergence of viral resistance through potent interference [24,26], selection pressures that limit escape [27], or activation of broadly protective host immunity [23]. In nature, DIPs arise as byproducts of error-prone viral replication, and the same mutational processes that generate viral diversity also drive diversity among DIPs. A key unanswered question is how ecological and coevolutionary interactions between viruses and their DIPs shape viral growth, disease severity, and viral persistence [28].

The potential evolution of viral resistance poses a significant challenge to DIP-based therapies. Early experimental work with VSV demonstrated that viruses can indeed evolve to escape DIP interference [29]. Time-shift experiments, where DIPs were tested against viruses from past, present, and future passages [30], revealed a dynamic of transient resistance [31]. The virus was resistant to contemporary DIPs but remained susceptible to those from the future or distant past. The specific mechanism driving this resistance was later identified as mutations in the viral polymerase that altered its binding specificity, rendering existing DIPs less effective [32].

Strategies to prevent this viral escape have been explored theoretically. A key concept is an "evolutionary conflict," where competing selection pressures make resistance a self-defeating strategy for the virus. For instance, within-host models for HIV show that any viral mutation that evades a TIP by reducing the production of a shared protein, such as the viral capsid that packages the viral genome, would simultaneously impair the virus's own replication fitness, creating a strong selective pressure against escape [33]. This principle was later shown to be robust at the population level, where additional evolutionary trade-offs further constrain the spread of resistant mutants [27]. A critical assumption underlying these frameworks, however, is that viral escape occurs through specific, discrete mutations against a static therapeutic agent. Therefore, investigating a "strong-mutation" environment, where a continuous flux of mutations allows for the evolution of both the virus and its interfering particles themselves, requires a different mathematical approach.

To investigate coevolutionary interactions between viruses and their DIPs, we build on phenotype-structured models that describe populations distributed over continuous trait spaces. An early formulation cast mutation as diffusion in phenotype space and selection through nonlocal interaction terms [34], providing a natural framework for strong-mutation regimes with intrinsically polymorphic populations. In contrast, classical adaptive dynamics assumes rare mutations and effectively monomorphic populations evolving through discrete trait substitutions [35], an approximation that may not hold in virus–DIP systems where mutations occur frequently within a single infection cycle. When mutations are small but frequent, selection–mutation equations admit asymptotic limits described by constrained Hamilton–Jacobi equations, with solutions concentrating as moving Dirac masses in phenotype space [36]. While these limits capture dominant trait trajectories, they may fail to represent the broad, heterogeneous distributions characteristic of rapidly mutating viral populations.

To address this limitation, alternative formulations retain finite mutation effects and track full phenotype distributions. Under diffusion and quadratic fitness landscapes, these models admit regimes in which trait distributions remain smooth (often Gaussian), allowing reduction to low-dimensional moment dynamics [37]. Related work has demonstrated traveling-wave or pulse solutions in coupled evolutionary systems driven by coevolutionary feedback [38]. Our framework builds on this class of models but introduces a distinct biological coupling: two interacting populations whose relationship is both ecological and generative, as defective interfering particles arise directly from the virus population they parasitize. This endogenous linkage creates a feedback that is missing in standard predator–prey or host–pathogen models, where species evolve independently. Consequently, selection emerges from the evolving overlap between virus and DIP phenotype distributions rather than from a fixed environment or external antagonist. This structure naturally produces sustained coevolutionary chase dynamics, providing a mechanistic bridge between phenotype-structured PDE theory and experimentally observed Red Queen patterns of transient resistance in virus–DIP systems.

Within this context, the Red Queen hypothesis (named after Lewis Carroll's character who declares, "it takes all the running you can do to keep in the same place") emphasizes that coevolution is driven by reciprocal rather than externally imposed selection [39]. In this view, antagonistic species must continuously adapt just to maintain their relative fitness.

One manifestation, the "Chase Red Queen," describes directional pursuit through multidimensional phenotype space: one population evolves to reduce its exploitable similarity to the other, while its antagonist evolves to close the phenotypic gap, leading to a persistent chase [40].

Building on these insights, we modify mathematical modeling of host-pathogen systems to explore virus–DIP coadaptation under strong mutation. We hypothesize that their dynamics may exhibit a Chase Red Queen pattern, with parameters such as mutation rate and interference strength determining whether the outcome is viral containment, escape, or extinction. By charting these coevolutionary regimes, we provide a theoretical basis for engineering TIPs that leverage trade-offs to maintain long-term viral control.

## Methods

We model DIP-virus coevolution in strong-mutation environments where interference strength varies with phenotypic distance between DIPs and viruses. Building on the framework of Alfaro et al. [38], we incorporate interactions across phenotype space and *de novo* DIP generation, capturing coevolutionary dynamics between phenotypically distant populations. We propose the following set of coupled PDEs to describe this system:

$$\partial_t v = \mu \nabla^2 v + \left( r_V - \alpha \|\mathbf{x}\|^2 - \beta \|\mathbf{x} - \bar{\mathbf{x}}_V(t)\|^2 - \eta - I_V[d](\mathbf{x}, t) \right) L(t) v - \gamma v \tag{1}$$

$$\partial_t d = \mu \nabla^2 d + \left( I_D[v](\mathbf{x}, t) d + \eta v \right) L(t) - \gamma d \tag{2}$$

All variables and parameters are summarized in Table 1. We describe each component of this system below.

The model in Eqs. 1 and 2 tracks the population densities of the virus, $v(\mathbf{x}, t)$, and DIP, $d(\mathbf{x}, t)$, as they evolve over a two-dimensional phenotype space, $\mathbf{x} \in \mathbb{R}^2$. We define this space abstractly to represent any set of continuous traits that could affect interference, rather than tying them to a specific biological mechanism. Examples could include affinity of virus or DIP genomic templates for the viral replicase or packaging proteins, or their replication rates. The evolution of these densities is driven by their phenotypic interactions, namely interference and competition for shared resources.

The terms $I_V[d](\mathbf{x}, t)$ and $I_D[v](\mathbf{x}, t)$ represent the total interference cost to the virus, $I_V$, and the corresponding benefit to the DIP, $I_D$:

$$I_V[d](\mathbf{x}, t) = \kappa_V (G * d)(\mathbf{x}) = \kappa_V \int_{\mathbb{R}^2} G(\mathbf{x} - \mathbf{x}') d(\mathbf{x}', t) \, d^2\mathbf{x}' \tag{3}$$

$$I_D[v](\mathbf{x}, t) = \kappa_D (G * v)(\mathbf{x}) = \kappa_D \int_{\mathbb{R}^2} G(\mathbf{x} - \mathbf{x}') v(\mathbf{x}', t) \, d^2\mathbf{x}' \tag{4}$$

$$G(\mathbf{x}) = \frac{1}{2\pi\sigma^2} e^{-\frac{\|\mathbf{x}\|^2}{2\sigma^2}} \tag{5}$$

The convolution integrals in Eqs. 3 and 4 capture the principle of phenotypic similarity: the closer DIPs are to the virus in phenotype space, the more strongly they interfere. In spatial population dynamics this formalism was developed to describe how organisms interact more strongly with nearby individuals than with distant ones [41]. Here, we extend the same idea into phenotypic space, where "distance" reflects trait differences rather than physical separation. The biological motivation for this principle is strong. For example, phenotypic divergence is known to reduce interference in VSV, where

**Table 1. Dimensional variables, parameters, and functions.**

| Symbol | Units | Description |
|---|---|---|
| **State variables** | | |
| $v(\mathbf{x}, t)$ | particles (phen. unit)$^{-2}$ | Virus density in phenotype space |
| $d(\mathbf{x}, t)$ | particles (phen. unit)$^{-2}$ | DIP density in phenotype space |
| $V(t)$ | particles | Total virus population |
| $D(t)$ | particles | Total DIP population |
| $\bar{\mathbf{x}}_V(t)$ | phen. unit | Mean virus phenotype (distribution centroid) |
| **Functions / operators** | | |
| $G(\mathbf{x})$ | (phen. unit)$^{-2}$ | Gaussian kernel |
| $\nabla^2$ | (phen. unit)$^{-2}$ | Laplacian operator for mutation (diffusion) |
| $I_V[d](\mathbf{x}, t)$ | time$^{-1}$ | Virus interference penalty |
| $I_D[v](\mathbf{x}, t)$ | time$^{-1}$ | DIP interference gain |
| $L(t)$ | dimensionless | Logistic population cap |
| **Parameters** | | |
| $r_V$ | time$^{-1}$ | Intrinsic viral replication rate |
| $\mu$ | (phen. unit)$^2$ time$^{-1}$ | Mutation rate (phenotypic diffusion) |
| $\alpha$ | time$^{-1}$ (phen. unit)$^{-2}$ | Strength of selection toward the origin |
| $\beta$ | time$^{-1}$ (phen. unit)$^{-2}$ | Strength of the aggregation penalty |
| $\eta$ | time$^{-1}$ | Rate of *de novo* DIP generation |
| $\kappa$ | (phen. unit)$^2$ particles$^{-1}$ time$^{-1}$ | Per-particle interference strength |
| $\gamma$ | time$^{-1}$ | Background removal (decay + dilution) rate |
| $\sigma$ | phen. unit | Standard deviation of the Gaussian kernel |
| $K$ | particles | Carrying capacity |

mutations to the viral replicase can abolish its ability to bind to DIP templates [32]. Likewise, the efficacy of DIP-mediated "genome stealing" in HIV-1 is determined by the similarity of their dimerization initiation sequences [33]. Our model abstracts this concept by using a Gaussian kernel, $G(\mathbf{x})$ (Eq. 5), to represent the interaction strength as a function of Euclidean distance in phenotype space. The strength of viral interference and DIP benefit are controlled by the parameters $\kappa_V$ and $\kappa_D$, and the rate at which interference declines with phenotypic distance is controlled by $\sigma$, the kernel's standard deviation.

We set $\kappa_V = \kappa_D \equiv \kappa$ to represent a zero-sum resource transfer. Following from the symmetry of Eqs. 3 and 4, total resources lost by the virus population are equal to the total resources gained by the DIP population, as shown by the integral equality:

$$\int_{\mathbb{R}^2} v(\mathbf{x}, t)\, I_V[d](\mathbf{x}, t)\, d^2\mathbf{x} = \int_{\mathbb{R}^2} d(\mathbf{x}, t)\, I_D[v](\mathbf{x}, t)\, d^2\mathbf{x}$$

(6)

Note that this conservation holds at the population level, not necessarily at a specific phenotype. We selected this zero-sum case as a neutral baseline because the net efficiency of resource hijacking is ambiguous. For instance, a DIP could be more efficient than a virus if its smaller genome enhances replication speed (a "positive-sum" outcome) [42]. Conversely, it could be less efficient if its deletions interfere with key functions like viral encapsidation or result in DIP self-interference (a "negative-sum" outcome) [43,44]. Therefore, a one-to-one transfer provides the clearest foundation for isolating the dynamics of competitive interference.

The experimental system that most directly realizes this assumption is a continuous-culture bioreactor (Fig 1), in which fresh host cells are supplied at a constant rate and culture fluid is removed to maintain a fixed volume [45,46]. In such a system, the steady-state cell density sets the carrying capacity $K$, the dilution rate sets the physical component of the removal rate $\gamma$, and the well-mixed condition eliminates spatial structure. For example, Frensing et al. implemented this design for continuous influenza virus propagation and observed periodic virus–DIP population oscillations consistent with the dynamics our model captures [10]. Serial passaging, the format used in the classical von Magnus [1] and DePolo et al. [31] experiments, represents a discrete analog in which dilution occurs in periodic pulses rather than continuously. We adopt the continuous formulation because it yields a mathematically tractable system, while noting that the qualitative dynamics persist under discrete transfer protocols [9,31].

For virus and DIP populations, $V$ and $D$, the logistic factor, $L(t)$, is given by:

$$L(t) = \left(1 - \frac{V(t) + D(t)}{K}\right) \tag{7}$$

$$V(t) = \int_{\mathbb{R}^2} v(\mathbf{x}, t)\, d^2\mathbf{x} \tag{8}$$

$$D(t) = \int_{\mathbb{R}^2} d(\mathbf{x}, t)\, d^2\mathbf{x} \tag{9}$$

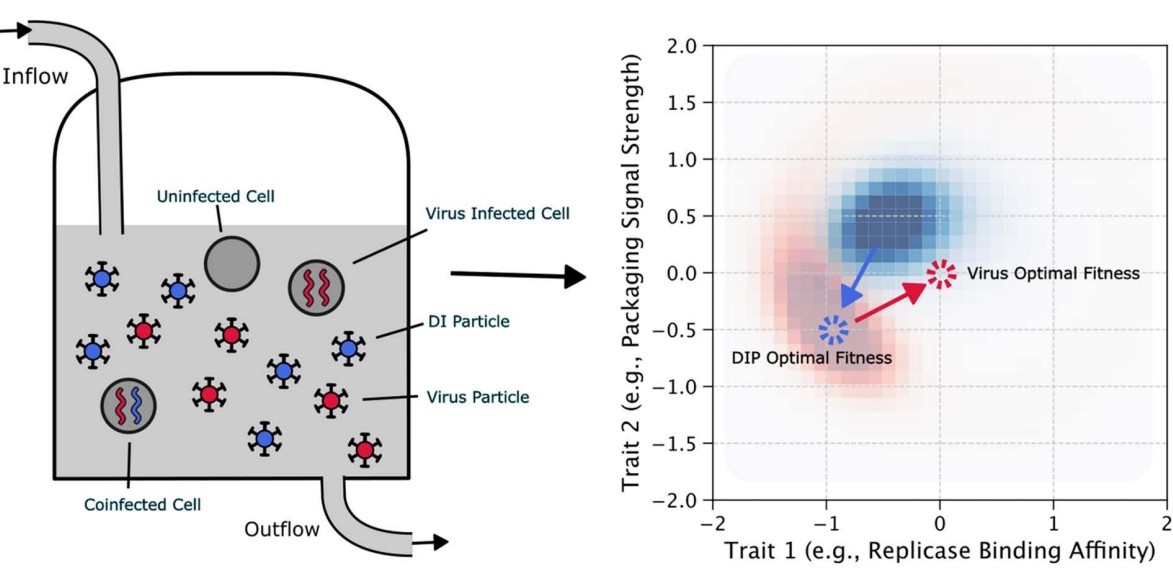

**Fig 1. Schematic representation of the virus–DIP coevolutionary system. Left panel:** Experimental setup modeled as a well-mixed bioreactor with continuous inflow of fresh cells and media, and outflow removing dead cells, viruses, and DIPs. Viral genomes (red) and defective viral genomes (blue) are packaged into particles that compete for cellular resources under finite carrying capacity constraints. **Right panel:** The diverse virus and DIP particles produced in the bioreactor can be represented as distributions in an abstract phenotype space, where their positions determine both intrinsic replication fitness and the strength of mutual interference. The virus phenotype optimum (red point) marks the location of its maximum intrinsic fitness at the origin, and the DIP phenotype optimum is co-located with the virus distribution (blue point). A central tradeoff explored in this study is whether the virus remains near its maximum intrinsic fitness despite strong interference from co-localized DIPs, or whether it moves to reduce interference, but at a cost to intrinsic fitness.

For simplicity, this formulation assumes that virus and DI particles contribute equally to saturating the carrying capacity $K$, though in reality, DIPs harboring shorter genomes may be less resource-intensive per particle.

We model the virus's intrinsic fitness landscape by adapting the framework of Alfaro et al. [38]. This landscape is composed of two quadratic penalty terms. The first, $-\alpha\|\mathbf{x}\|^2$, establishes a fixed fitness optimum at the origin, penalizing viruses for deviating from this global peak. The second is an aggregation term, $-\beta\|\mathbf{x} - \bar{\mathbf{x}}_V(t)\|^2$, which penalizes viruses for straying from the population's current mean phenotype, $\bar{\mathbf{x}}_V(t)$. The quadratic form is chosen for the generic local approximation, as the leading-order term in a Taylor expansion of any smooth fitness landscape near an isolated optimum. The mean phenotypes of the virus and DIP populations are calculated as:

$$\bar{\mathbf{x}}_V(t) = \frac{1}{V(t)} \int_{\mathbb{R}^2} \mathbf{x}\, v(\mathbf{x}, t)\, d^2\mathbf{x} \tag{10}$$

$$\bar{\mathbf{x}}_D(t) = \frac{1}{D(t)} \int_{\mathbb{R}^2} \mathbf{x}\, d(\mathbf{x}, t)\, d^2\mathbf{x} \tag{11}$$

The DIP mean phenotype $\bar{\mathbf{x}}_D(t)$ does not appear in the governing equations but is used throughout our analysis to quantify coevolutionary dynamics.

Mathematically, the aggregation term ensures population cohesion and prevents non-biological ring structures and fragmentation [38]. Viral populations are understood to function as cohesive "quasispecies," existing not as a single genotype but a cloud of related mutants centered around a dominant "master copy" [47]. The ensemble provides complementation benefits that individual clones lack [48]. A virus that becomes phenotypically isolated from this swarm loses these benefits, incurring a fitness cost. Therefore, the $\beta$ term models a plausible selective pressure that penalizes phenotypic outliers, promoting the cohesion that may be necessary for the quasispecies to act as a collective unit of selection. DIPs are obligate parasites that do not share resources with one another so there is no selective pressure favoring DIP cohesion.

We also account for the *de novo* generation of DIPs from replication errors in the virus population. These errors produce a spectrum of outcomes. Some errors produce viable virus variants whose small phenotypic shifts are captured by our diffusion term ($\mu\nabla^2 v$). Other errors can be lethal, creating non-viable particles that cannot replicate at all and are thus excluded from our model. To model the *de novo* generation of DIPs we focus on the critical intermediate class of replication errors: defective genomes that are replication-incompetent on their own but retain the necessary sequences to be replicated and packaged by the machinery of the standard virus. This generation is modeled as a transfer of density from the virus to the DIP population at a rate $\eta$. A key assumption of this term is that a newly generated DIP inherits the exact phenotype of its parent virus. We ground this assumption in the requirement that for a DIP to be conditionally replication-competent, it must retain significant functional similarity to its parent. Major phenotypic changes would likely render an emergent DIP non-viable and thus irrelevant to the coevolutionary dynamic. This assumption therefore focuses the model on the subset of *de novo* DIPs that are most likely to interfere.

Mutation is modeled as diffusion in phenotype space, analogous to the process of diffusion of particles in a physical space. The random motion is caused by the random changes to phenotype by genetic mutations. An initially clonal population will therefore "diffuse" or spread out across the phenotype space over time. We mathematically represent this with the Laplacian operator, $\nabla^2 = \partial^2/\partial x_1^2 + \partial^2/\partial x_2^2$ [49]. For simplicity, we assume this diffusion is isotropic and that both the virus and DIP populations share the same diffusion coefficient, the mutation rate parameter, $\mu$.

We define the intrinsic viral growth rate as $r_v$. We then assume particles can be removed from the population in two ways. The first is intrinsic biological decay, such as the natural degradation or inactivation of viral particles [50]. The second is physical removal from the system, corresponding to the continuous dilution in experimental setups like the constant-volume bioreactor (Fig 1). In such a system, this physical dilution rate is defined as the outflow rate divided by

the constant fluid volume of the bioreactor. For simplicity, our model combines these biological and physical removal mechanisms into a single constant rate, $\gamma$, applied equally to both virus and DIP populations.

To reduce the number of free parameters, we rescale time by the intrinsic viral replication rate $r_V$ and phenotypic space by the kernel width $\sigma$, so that time is measured in units of the viral replication cycle and phenotypic distance in standard deviations of the Gaussian kernel:

$$\tilde{t} = r_V t, \qquad \tilde{\mathbf{x}} = \frac{\mathbf{x}}{\sigma},$$
$$\tilde{v}(\tilde{\mathbf{x}}, \tilde{t}) = \sigma^2 v(\mathbf{x}, t), \qquad \tilde{d}(\tilde{\mathbf{x}}, \tilde{t}) = \sigma^2 d(\mathbf{x}, t),$$

The carrying capacity $K$ retains its dimensional value in particles for biological interpretability. This transformation yields a system identical in form to Eqs. 1–2 with $r_V = \sigma = 1$. The full dimensionless equations, rescaled variables, and parameter definitions are given in S1 Text. Then, we drop tildes and work exclusively in dimensionless variables; all parameter values reported in the Results are these dimensionless groups.

We define the summary statistics used throughout the Results. Phenotypic distance is the Euclidean distance between the centroids of the virus and DIP distributions:

$$\Delta(t) = \|\bar{\mathbf{x}}_V(t) - \bar{\mathbf{x}}_D(t)\| \tag{12}$$

To compare virus susceptibility across time points independently of population size, we define a normalized interference between a virus snapshot at time $t_i$ and a DIP snapshot at time $t_j$:

$$I(t_i, t_j) = \kappa \int_{\mathbb{R}^2} \hat{v}(\mathbf{x}, t_i) \left( G * \hat{d} \right)(\mathbf{x}, t_j) \, d^2\mathbf{x} \tag{13}$$

where $\hat{v} = v/V$ and $\hat{d} = d/D$ are the population-normalized densities. This quantity measures the per-capita interference a virus with the phenotypic profile of time $t_i$ would experience from DIPs with the phenotypic profile of time $t_j$, independent of population sizes. For a given DIP snapshot at time $t_j$, the fold resistance of the virus at time $t_i$ is:

$$R(t_i, t_j) = \frac{\max_{t_k} I(t_k, t_j)}{I(t_i, t_j)} \tag{14}$$

A fold resistance of 1 indicates maximal susceptibility, while higher values indicate greater resistance. This measure is analogous to the fold-resistance assays used in experimental time-shift studies [31], where viral resistance is quantified relative to the most susceptible strain.

We formally state the initial-boundary value problem that forms the basis of our analysis. The goal is to find the non-negative population densities $v(\mathbf{x}, t)$ and $d(\mathbf{x}, t)$ satisfying Eqs. 1 and 2. For our numerical implementation, this system is solved on a bounded rectangular domain $\Omega \subset \mathbb{R}^2$ subject to homogeneous Neumann (zero-flux) boundary conditions $\frac{\partial v}{\partial \mathbf{n}} = 0$ and $\frac{\partial d}{\partial \mathbf{n}} = 0$ on the boundary $\partial\Omega$ where $\mathbf{n}$ is the outward normal vector. The system evolves from prescribed initial population distributions $v(\mathbf{x}, 0) = v_0(\mathbf{x})$ and $d(\mathbf{x}, 0) = d_0(\mathbf{x})$. All code is publicly available at https://github.com/shivmuthupandiyan/virus-dip-coevolution

## Use of artificial intelligence tools

Large language model AI tools (OpenAI ChatGPT and Anthropic Claude) were used during this study to assist with writing and debugging simulation code and generating initial data visualizations. All AI-generated code was reviewed, tested, and

validated by the authors before use. The mathematical model formulation, analysis, interpretation of results, and all scientific conclusions presented in this article are entirely the authors' own work.

## Results

### Numerical approach

To illustrate the model's core dynamics, we solved the system numerically on a [-5, 5] × [-5, 5] domain with zero-flux boundary conditions. Unless otherwise specified, we used the following baseline parameters: $\mu = 10^{-3}, \eta = 10^{-3}, \alpha = 0.025, \beta = 0.025, \kappa = 10^{-7}, \gamma = 0.3, K = 10^8$. Simulations were initialized with a virus population of $10^6$ and a DIP population of either 0 or $10^6$. The initial virus and (when present) DIP populations were distributed as Gaussians (standard deviation = 0.4) centered at (1, 1) and (1.4, 0), respectively. A population was considered extinct when its total size fell below $10^2$.

Unless otherwise noted, baseline simulations were run to $T = 200$. References to "stable" or "sustained" dynamics throughout the Results describe behavior observed over the second half of the simulation window ($t \in [100, 200]$); we do not claim formal mathematical stability. At typical viral replication times of 5–9 hours [51], $T = 200$ corresponds to 40–70 days of continuous culture, comparable to the duration of longer continuous culture experiments [10]. Extending simulations further would resolve additional transient oscillations near regime boundaries, but would model timescales beyond those of most experimental systems. To assess sensitivity to simulation duration, we compared outcome classifications at $T = 200$ and $T = 400$ (S1 Table). Overall concordance was 93%. The primary discrepancy was at the chase–coexistence boundary: 23% of chase classifications transitioned to coexistence by $T = 400$. Because these long-lived transient oscillations persist for hundreds of viral generations, exceeding typical experimental observation windows, the chase prevalence reported below includes both sustained and slowly damping coevolutionary dynamics.

**Parameter selection and biological grounding.** Our baseline parameters were chosen to place the model in a biologically plausible regime, informed by available experimental measurements where possible. We briefly justify each choice. Further validation of the parameters can be found in the discussion.

The carrying capacity, $K$, was set to $10^8$ to give biologically reasonable population sizes [52]. The decay rate $\gamma = 0.3$ implies that particle removal occurs at roughly 30% of the intrinsic replication rate. For a typical mammalian RNA virus (e.g., picornaviruses or alphaviruses), the intracellular replication time is on the order of 5–9 hours [51]. This corresponds to an effective particle half-life in culture media of approximately 15–20 hours, which is in the range of measured stability values [53]. Additionally, in a bioreactor context, dilution rate is an experimentally tunable parameter that directly sets this value.

The *de novo* DIP generation rate $\eta = 10^{-3}$ implies that roughly one in a thousand replication events produces a defective genome capable of conditional replication. Stauffer Thompson and colleagues estimate an effective generation rate of "interference equivalents", a proxy for DIPs, in high-MOI environments, at $10^{-3}$ per cell [43]. Additionally, deep sequencing studies of RNA virus populations detect baseline levels of defective viral genomes at frequencies on the order of $10^{-4}$–$10^{-3}$ relative to standard genomes, depending on conditions, before quickly enriching [54,55].

The mutation rate parameter $\mu = 10^{-3}$ controls the rate of phenotypic diffusion. RNA virus mutations tend to be in the range of $10^{-6}$–$10^{-4}$ substitutions per nucleotide per replication round [56]. However, translating this into a diffusion coefficient in an abstract phenotype space requires assumptions about the genotype-phenotype map that are not currently available. We therefore treat $\mu$ as an uncertain estimate, and note that the sensitivity analysis below shows negligible dependence on $\mu$, indicating a robustness to this uncertainty.

The interference strength $\kappa = 10^{-7}$ implies that DIP populations on the order of $10^7$ are required to produce strong interference effects. This is consistent with the biological requirement for coinfection. At a carrying capacity of $K = 10^8$ particles and typical RNA virus burst sizes of $10^1$–$10^3$ particles per infected cell [57], the implied host-cell population is on the order of $10^5$–$10^7$ cells. A DIP population of $10^7$ in such a culture corresponds to a multiplicity of DIP (MODIP) of roughly $1 - 10$

per cell, which falls in the range where dose-dependent interference transitions from weak to strong [43,58,59]. At lower DIP densities, coinfection is too infrequent for interference to materially affect viral fitness.

The fitness landscape parameters $\alpha$ = 0.025 and $\beta$ = 0.025 are the most difficult to ground empirically, as they refer to the curvature of an abstract fitness surface. The value of $\alpha$ implies that a virus displaced by one standard deviation from the fitness optimum suffers a 2.5% decrease in intrinsic growth rate. This is qualitatively consistent with measures of fitness costs associated with mutations in RNA viruses, ranging from negligible to 10–20% per mutation [60,61]. The aggregation parameter $\beta$ imposes an equivalent penalty for distance from the phenotypic mean, reflecting qualitative fitness advantages of quasispecies cohesion, although precise quantitative measures of the cost of isolation are highly system-dependent [47,48,62].

**Baseline dynamics without interference.** First, we consider a baseline scenario where virus infection produces non-interfering deletion variants as byproducts of viral replication; here the interference strength $\kappa$ is set to zero (Figs 2A–2C). These particles are not DIPs because they do not interfere with the virus. In this case, evolution of the virus population is driven solely by the intrinsic fitness landscape, and its mean phenotype moves toward the optimum at the origin. These mutants are generated from the virus at a rate $\eta$ but cannot replicate on their own because the interference benefit $I_D$ is zero. Consequently, the deletion mutant population is simply "dragged along" by the virus, remaining co-localized in phenotype space. Its population stabilizes when *de novo* generation ($\eta V$) is balanced by decay ($\gamma d$), yielding a population ratio of $D = \eta V/\gamma$. Without interference, there is no coevolutionary pressure, and the mutants act only as a minor sink on

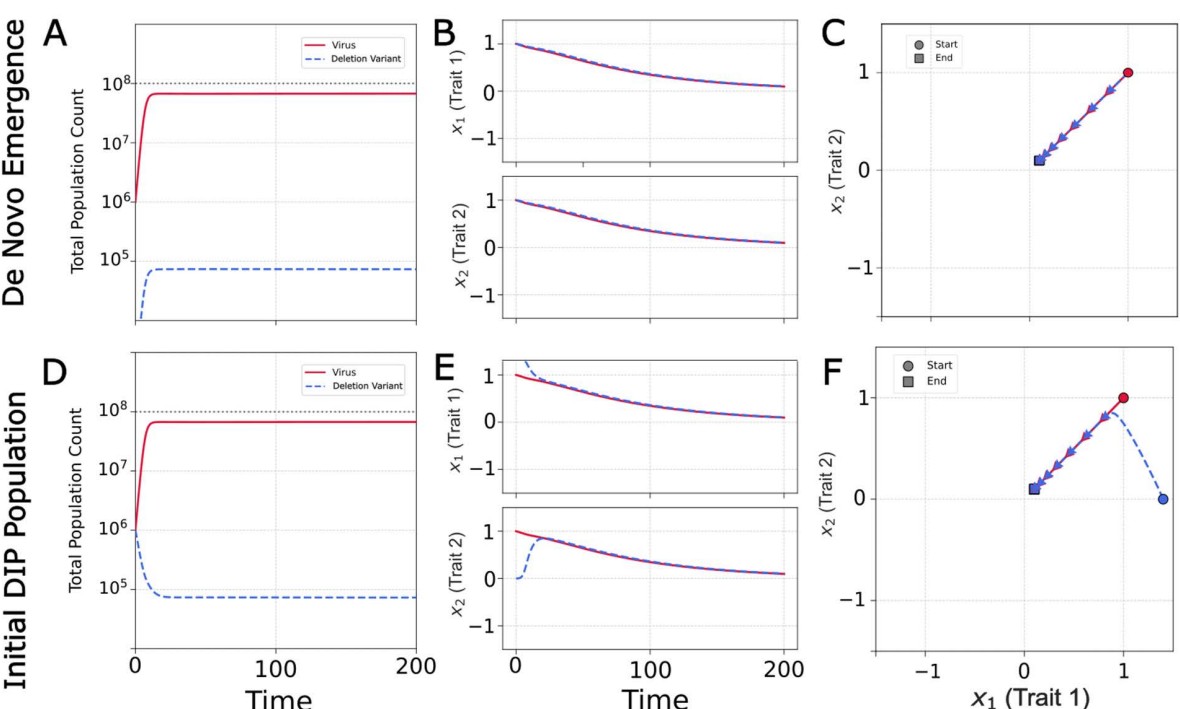

**Fig 2. Virus population and evolutionary dynamics in the absence of interference by variants ($\kappa$ = 0, $\mu$ = 10$^{-2}$).** (A) Baseline scenario with a deletion variant (blue dash) population emerging *de novo* from the virus (solid red) population. Total population sizes over time show viral dominance with deletion variant stabilizing at low levels. (B) Individual $x_1$, $x_2$ plots show both populations evolving toward the virus fitness optimum at the origin. (C) 2D phenotype trajectory demonstrates simple linear movement toward the optimum with virus and deletion variant populations remaining co-localized. (D) Robustness test with deletion variant populations initially phenotypically distant from the virus. Total population sizes converge to the same lowered level. (E) Deletion variant phenotypes converge to virus, while virus converges to origin. (F) 2D trajectory shows virus trajectory to the origin and deletion variant trajectory to the virus.

the virus population. This can map to a situation often observed during low multiplicity of infection (MOI) passages *in vitro*, where the few deletion mutants that are generated are not ever able to significantly establish themselves, causing minimal suppression of viral titers [63]. Here, the zero interference corresponds to passaging with sufficiently low MOI that the DIPs never coinfect the virus-infected cells, preventing any interference.

If the simulation begins with a non-interfering mutant population that is separate from the virus in phenotype space (Figs 2D–2F), the initial population of mutants simply decays due to its inability to steal from the virus and replicate. It is steadily replaced by a *de novo* population that emerges co-localized with the virus. The phenotype centroid slowly shifts towards the virus centroid as more deletion mutants emerge from the virus, eventually driving the system back to the same steady state at the origin.

**Interference drives coevoluationary chase.** When interference is introduced ($\kappa > 0$), the dynamics become coevoluationary. DIPs can now replicate by parasitizing phenotypically similar viruses, creating a new selective pressure that disfavors viral phenotypes close to the DIPs. In a simulation where DIPs emerge *de novo* (Figs 3A–3C), the virus experiences conflicting evolutionary forces: a directional selection towards the origin (the $\alpha$ term) and a selection away from the growing DIP population (the $\kappa$ term). Without a fixed fitness peak ($\alpha = 0$), the model would produce an "Escalatory Red Queen" dynamic, an endless arms race across a phenotype dimension. However, we assume that mutations that evade DIPs typically incur costs to intrinsic viral fitness, a phenomenon that has been observed in cases of antiviral and immune system escape [62]. The $\alpha$ term formalizes this constraint by establishing a global fitness optimum. This conflict results in a transient chase dynamic. As the virus nears the origin, the DIPs it generates become dense enough to repel it, causing the mean phenotype to overshoot the optimum before the $\alpha$ penalty pulls it back. While the trajectory of the population centroid is relatively simple, it arises from complex changes in the underlying distributions as they navigate this dynamic fitness landscape.

Sustained coevoluationary chase dynamics emerge when the initial virus and DIP populations are phenotypically separated (Figs 3D–3F). The initial offset breaks the linear trajectory seen previously, inducing a circular pursuit: viruses are selected to move away from the DIPs, while DIPs are selected to better track the viruses. The $\alpha$ term acts as a centripetal force, pulling the populations toward the origin and converting a linear arms race into a stable, cyclic pursuit. This dynamic, a chase across a multidimensional phenotype space, is the definition of a Chase Red Queen [40]. (See S1 Video)

The separated initialization models the practically relevant scenario of introducing a DIP or therapeutic interfering particle into a culture with an established virus population. In such cases, the introduced particle would not be expected to match the contemporary virus phenotype, whether it arose from a different passage history or was engineered independently [24]. Together, the two initializations bracket the relevant extremes: *de novo* emergence (co-localized, Figs 3A–3C) and deliberate introduction of a pre-existing interfering particle (separated, Figs 3D–3G).

**Chase dynamics are reflected in both population size and phenotype.** The population oscillations in Fig 3A and 3D are a consequence of interference (von Magnus effect) that drives a predator-prey feedback observed in laboratory cell cultures of RNA viruses [5,7,8] as well as in mice [6]. High levels of DIPs suppress the virus; this depletes the resources available for DIP replication, causing the DIP population to fall; the virus population is then able to recover, restarting the cycle. Our model extends this dynamic by allowing the strength of virus–DIP interference to depend both on phenotypic distance and on the population sizes of $V$ and $D$ via the interference terms of Eqs. 4 and 3, respectively, with phenotypic distance itself oscillating as the virus escapes and the DIP gives chase. The chase dynamics, continuous cycles of viral escape and DIP pursuit, directly influence population sizes. These cycles are not always synchronized: viral recovery can occur either after a DIP crash or by creating sufficient phenotypic distance to weaken interference, linking population cycles to evolutionary dynamics.

Snapshots of the population densities reveal the coevoluationary chase in detail (Fig 3G). At any given time, both the virus and DIP populations exist not as single points but as diffuse clouds in phenotype space. This distribution

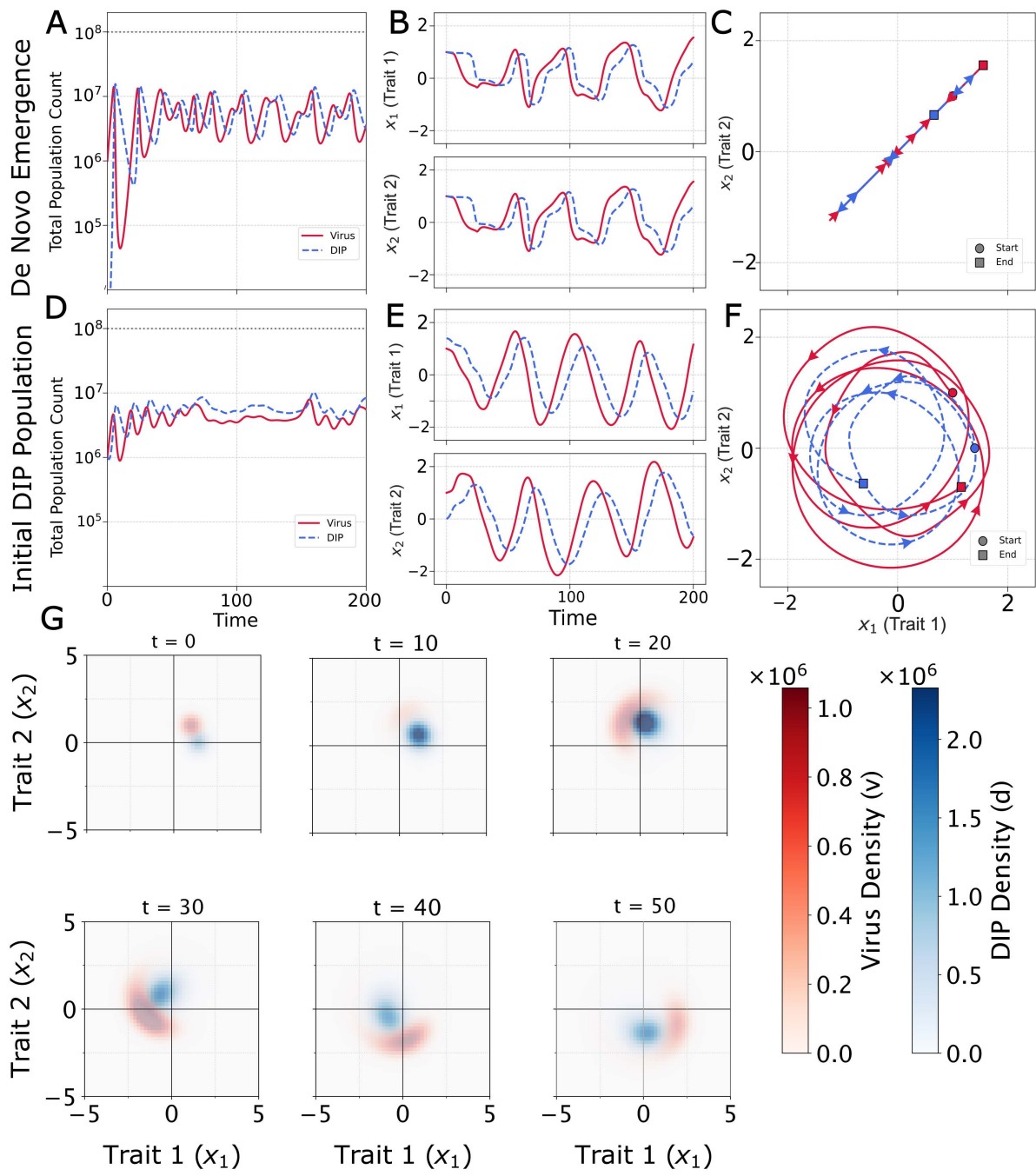

**Fig 3. Population and evolutionary dynamics with virus–DIP interference ($\kappa > 0$, $\mu = 10^{-2}$). (A)** Coevolutionary dynamics when DIPs (striped blue) emerge *de novo* from the virus (solid red) population. Oscillatory population dynamics demonstrate the von Magnus effect. **(B)** Phenotype trajectories show conflicting evolutionary pressures as the virus experiences a pull toward the fitness optimum ($\alpha$ term) and a push away from interfering DIPs ($\kappa$ term), resulting in overshoot dynamics. **(C)** Two-dimensional trajectory shows the oscillation along a line as the virus attempts to balance intrinsic fitness gains with interference avoidance (See S1 Video). **(D)**. Initial phenotypic separation induces a continuous coevolutionary pursuit where viruses evolve to escape DIP interference while DIPs track the moving viral target. Population oscillations reflect the dynamic strength of interference as phenotypic distance varies. Note that the population oscillations in (D) reflect transient dynamics that may not have fully equilibrated. **(E)** Large-amplitude oscillations in both phenotype dimensions demonstrate active coevolutionary chase. **(F)** Chase Red Queen trajectory in phenotype space. **(G)** Heatmaps of virus (red) and DIP (blue) densities at six sequential time points from the simulation in Figs 3D–3F, showing the coevolutionary chase as it unfolds. Each population exists as a diffuse cloud in phenotype space, analogous to a viral quasispecies. Over time, the DIP cloud tracks the virus cloud as it moves to evade interference, with the two populations orbiting through phenotype space in a cyclic pursuit (see S2 Video).

is analogous to a viral quasispecies, where a central dominant phenotype is surrounded by a spectrum of mutants. The heatmaps illustrate the continuous pursuit, with the DIP cloud tracking the virus cloud as it moves to evade interference.

The sustained chase dynamic shown in Fig 3 represents one of four possible outcomes. Depending on the model parameters, the coevolutionary interaction can also lead to DIP extinction, virus–DIP coextinction, or a stable coexistence with no oscillations. There is no region for virus extinction alone, as DIPs cannot replicate in the absence of virus, so virus extinction entails DIP extinction. The next section explores how key parameters determine which of these regimes will emerge.

## Sensitivity analysis

We performed a global sensitivity analysis using Latin Hypercube Sampling (LHS) (n = 10,000) to explore how variation in six key parameters shapes the prevalence of distinct coevolutionary regimes. Unlike one-at-a-time sensitivity analysis, which varies a single parameter while holding all others fixed, LHS draws each sample from the full six-dimensional parameter space simultaneously, with a stratified design to ensure uniform coverage [64]. The resulting stacked proportion plots (Figs 4A–4F) reveal how the proportion of simulations in each regime changes as individual parameters vary, yet still robust to simultaneous uncertainty in all other parameters, not conditional on any particular fixed values.

For each parameter, outcomes were classified as Chase, Coexistence, DIP Extinction, or Coextinction. We swept parameter ranges: $\kappa : (10^{-1}, 10^2)$, $\eta : (5 \times 10^{-4}, 5 \times 10^{-1})$, $\alpha : (3 \times 10^{-3}, 3)$, $\beta : (3 \times 10^{-3}, 3)$, $\gamma : (0, 0.9)$, $\mu : (10^{-4}, 10^{-2})$, all on log scales except for $\gamma$. These ranges were selected to span at least two orders of magnitude around the baseline values, encompassing biologically plausible dynamics while avoiding degenerate regimes. Each parameter was sampled from a uniform distribution on its respective scale (log-uniform for log-scaled parameters, uniform for $\gamma$). Because the system is continuous and the dynamics depend only on the ratios $V/K$ and $D/K$, initializing at $10^{-2}$ with $K = 1$ is equivalent to initializing at $10^6$ with $K = 10^8$. This allows population sizes to be read directly as fractions of carrying capacity, simplifying interpretation across the parameter sweep.

All simulations were run to a fixed end time of $T = 200$ (in units of the viral replication cycle). To minimize the influence of transient dynamics on outcome classification, we assessed outcomes using only the second half of each simulation ($t \in [100, 200]$). Populations were initialized at $10^{-2}$ particles for both virus and DIP, with carrying capacity at 1. A simulation was classified as *Coextinction* if the virus population fell below $10^{-6}$ at any point (triggering early termination) or remained below $10^{-6}$ at $T = 200$, and as *DIP Extinction* if the DIP population fell below $10^{-6}$ while the virus persisted. For surviving populations, we computed the Euclidean distance $\Delta(t)$ (Eq. 12) between virus and DIP distribution centroids at each time point over the assessment window and classified the outcome as *Chase* if the standard deviation of $\Delta(t)$ exceeded 0.1, indicating sustained oscillatory pursuit, and as *Coexistence* otherwise.

Several robust patterns emerge. Interference strength ($\kappa$, Fig 4A) shows a clear progression: when $\kappa$ is low, nearly all simulations result in coexistence, as interference is too weak to generate selective pressure. At intermediate $\kappa$, chase dynamics become possible (~20% of outcomes), but excessively strong interference pushes the system toward coextinction. The *de novo* DIP generation rate ($\eta$, Fig 4B) has an equally strong effect: at low $\eta$, DIPs fail to persist and chase is common, but higher values reduce the opportunity for phenotypic separation by continuously regenerating DIPs near the virus, enforcing coexistence.

The phenotypic sensitivity parameters ($\alpha$, Fig 4C; $\beta$, Fig 4D) both display similar trade-offs. Low values favor chase by allowing viruses to maintain phenotypic separation, while higher values suppress coevolution and shift the system toward coexistence and extinction outcomes. This symmetry suggests that both parameters act as modulators of how tightly viral and DIP dynamics are coupled.

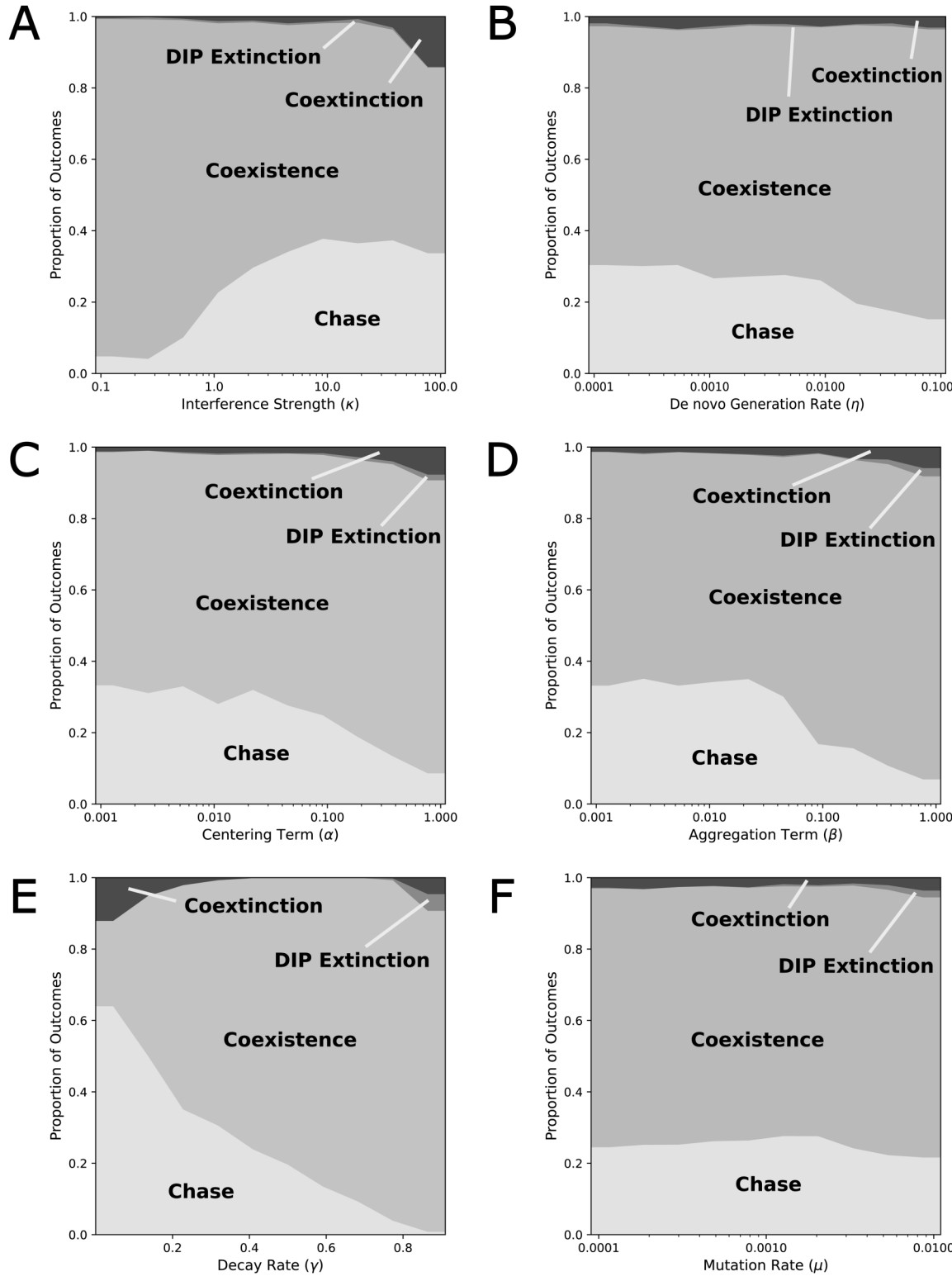

**Fig 4. Sensitivity analysis of the model. (A) Interference strength** ($\kappa$). Increasing the interference strengths of DIPs burdens the virus, shifting populations from coexistence towards chase regimes, until interference is so strong that viruses start to go extinct. **(B)** *De novo* **generation** ($\eta$). Larger *de novo* emergence of DIPs colocalized with virus slightly favor coexistence over chase outcomes. **(C) Centering Term** ($\alpha$). Drawing the virus closer to its maximum fitness (origin), suppresses its ability to escape, favoring coexistence and extinction outcomes. **(D) Aggregation Term** ($\beta$). Increasing the

aggregation discourages virus phenotypic diversification, as mutations that drive diversification are disfavored. Effects mirror those of the centering term. **(E) Decay Rate** ($\gamma$). At very low decay rates, virus and DIP growth approach their carrying capacity because there is very little dilution, leading to chase outcomes and coextinction. As the decay rate increases, variation is selected against more strongly, preventing a chase from occurring, until very high decay rates, when DIP extinction and coextinction rise again. **(F) Mutation rate** ($\mu$). Mutation rates have a negligible effect in the examined parameter range, as the selection is much stronger in the mutation-selection balance.

The dilution rate ($\gamma$, Fig 4E) exhibits the most complex effects. At low values, chase is highly prevalent (nearly 50%) alongside elevated extinction rates, reflecting that limited dilution disproportionately benefits DIPs and forces the virus into rapid adaptation or collapse. By analogy with a bioreactor, allowing the dilution rate to approach zero means the influx of fresh cells for infection or coinfection approaches zero, and the system approaches a closed system, as total population approaches the carrying capacity limit and the logistic factor $L$ approaches zero. As $\gamma$ increases, extinction decreases and coexistence grows more common, indicating that faster removal weakens DIP pressure. Higher dilution harms DIP populations directly, by removal, and indirectly, by reducing viruses to coinfect with, weakening DIPs relative to viruses. However, at the highest values, strong dilution suppresses both populations and again increases extinction. This reveals that $\gamma$ tunes the relative balance of viral escape and DIP persistence in a nonlinear manner.

Mutation rate ($\mu$, Fig 4F) has comparatively weaker effects: chase maintains a consistent prevalence of ~25%, while coexistence dominates (~70%). DIP extinction and coextinction occur only rarely in this range. The relatively low frequency of chase here reflects the chosen parameter bounds; other ranges could yield a different balance.

Overall, the LHS sensitivity analysis shows that chase dynamics are most common under intermediate interference strength, low-to-moderate DIP generation, and low dilution. Coexistence dominates large regions of parameter space, especially when interference is weak, mutation rate is moderate, or dilution is strong. Extinction outcomes emerge primarily at the extremes of parameter values. Together, these results highlight that coevolutionary chase requires a narrow balance of forces. Coexistence is the dominant outcome across broad regions of parameter space, and extinction arises predictably when any parameter is pushed to an extreme that overwhelms one or both populations.

## Cyclic escape

Our model's coevolutionary chase dynamics provide a mechanistic explanation for classic experimental observations of cyclic viral resistance. Although virus and DIP populations only ever interact contemporaneously during infection, time-shift experiments, in which historical isolates are crossed against a fixed DI particle stock, are a standard diagnostic tool for distinguishing modes of coevolution [30]. In such experiments, DePolo et al. found that VSV is most resistant to DIPs from the recent past, yet remains susceptible to DIPs from the distant past and future [31]. This pattern of transient resistance argues against a simple Escalatory Red Queen arms race, where resistance to past DIPs would be expected to increase monotonically. Instead, it is a hallmark of a Chase Red Queen dynamic, which our model conceptualizes as a cyclic pursuit within a constrained phenotype space. Because the evolutionary trajectory is orbital, a virus population at a given time can be phenotypically closer to DIPs from the distant past or future than to the DIPs from the recent past it has just evolved to escape. Our model reproduces this pattern using the normalized interference and fold-resistance measures defined in Eqs. 13 and 14.

We first generated a heatmap showing how viral resistance to DIP interference varied over time (Fig 5A). The diagonal stripes reveal alternating cycles of high and low resistance, indicating ongoing coevolution between the virus and DIP populations. We then fixed one DIP population (from time T = 77) and measured its interference against viruses from surrounding passages (Fig 5B). The virus was least resistant to DIPs from its own time, more resistant to those from the recent past—having just evolved to escape them—and less resistant again to those from the distant past or future. Finally, to compare these predictions with experimental data, we devised a time-shift sampling approach to mimic the DePolo

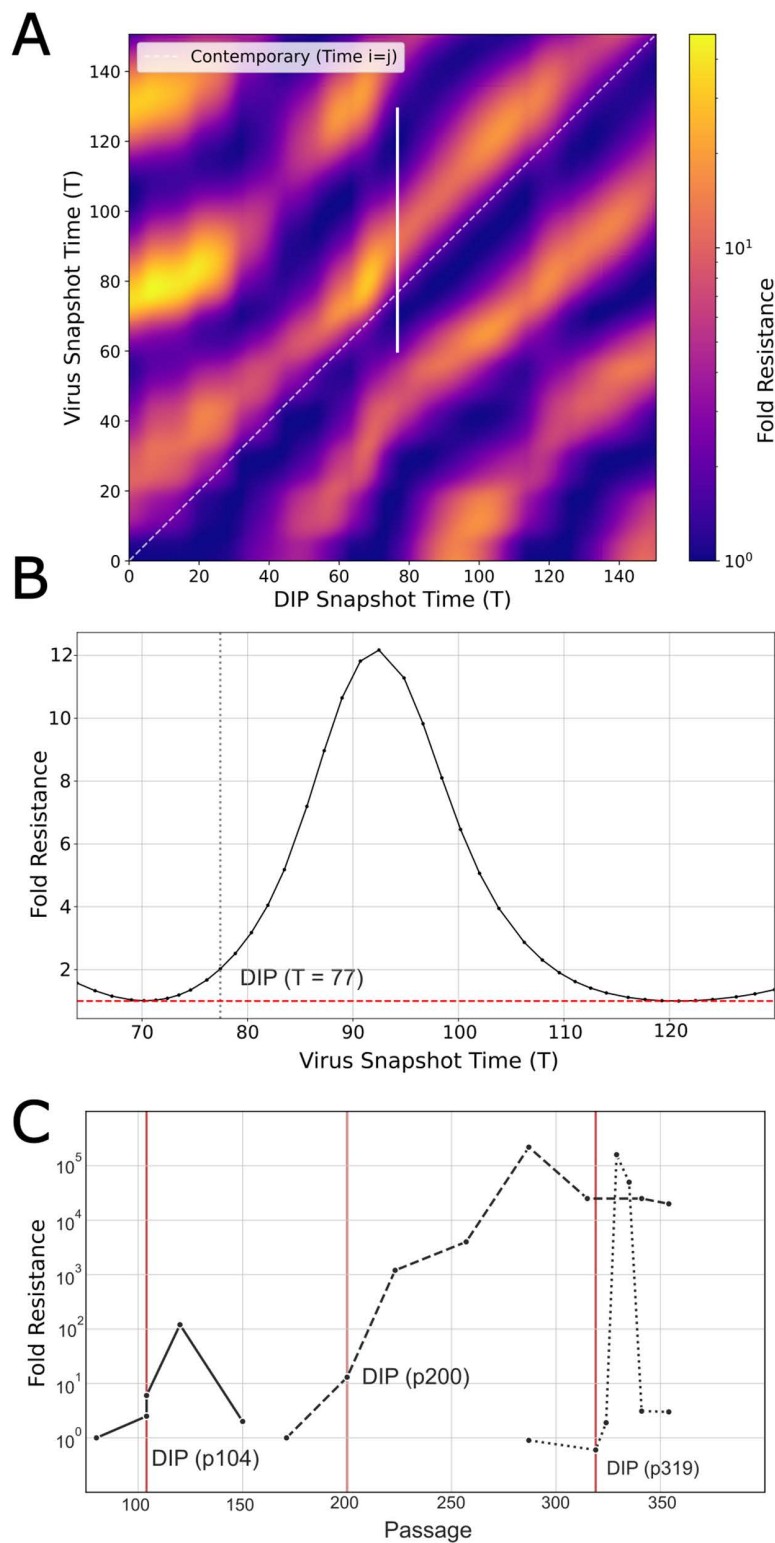

**Fig 5. Cross passage viral resistance. (A)** Resistance of any given virus passage (y-axis) against any given DIP passage (x-axis), measured as fold resistance. The white diagonal represents the resistance against contemporary passage (DIP Snapshot Time $T_i$ = Virus Snapshot Time $T_j$). The diagonal stripes represent a coevolutionary chase where viruses are most resistant to DIPs from the recent past, before losing their resistance. The vertical white

line shows the subset of resistances depicted in **B. (B)** Example viral resistance curve against a specific DIP (T = 77), showing a transient increase in resistance for several passages, before reverting to the baseline resistance (red dashed line). The domain corresponds to the vertical white line in **A. (C)** Replotting of data from DePolo et al. (1987), showing viral resistance to DIPs cycling over several passages [31]. In the original experiments, virus clones were isolated from specific passages in a serial undiluted passage series of VSV, frozen, and later tested against a fixed DI particle preparation (isolated at passage 200) by measuring virus yield suppression as a function of DI particle input multiplicity. Fold resistance was defined as the ratio of DI particles required to achieve equivalent suppression of a test virus relative to the most sensitive strain.

serial transfer experiments (Fig 5C). The resulting pattern matched the transient cycles of resistance observed in the original study.

## Evolutionary conflict

Our model also provides a framework to examine "evolutionary conflict," an emerging concept in the design of therapeutic interfering particles (TIPs). In a multi-scale model of HIV, Rast et al. (2016) proposed that TIPs could create an evolutionary trap: any viral mutation that evades interference also incurs a significant fitness cost, blocking the evolution of resistance [27]. In our model, this constraint is captured by the $\alpha$ parameter, which penalizes deviations from the fitness optimum at the origin.

This raises a key question: under what conditions does the selective pressure to escape TIPs ($\kappa$) outweigh the fitness cost of deviating from the optimum ($\alpha$)? To probe this trade-off, we examined virus population sizes and virus–DIP phenotypic distances across a range of $\kappa$ and $\alpha$ values with all other parameters at their baseline values and simulations run to $T = 100$ (Fig 6). Summary statistics were computed over the final 20% of each simulation ($t \in [80, 100]$) to reduce sensitivity to transient dynamics. The "final virus population" (Fig 6A) is the time-averaged total virus population over this window, normalized by carrying capacity and plotted on a logarithmic scale. The "distance from optimal fitness" is the time-averaged Euclidean distance between the virus centroid and the fitness optimum at the origin, $\|\bar{x}_V\|$, over the same window.

As expected, virus populations decline with increasing interference by DIPs, largely independent of $\alpha$ (Fig 6A). However, the average phenotypic distance between virus and DIP centroids reveals a clear threshold behavior (Fig 6B). Fig 6C illustrates several cases. When the fitness penalty for deviation is strong (e.g., $\alpha \geq 0.5$), the virus remains "trapped" at the origin regardless of the interference strength. Any mutation away from the optimum is so costly that the virus cannot stably evolve any DIP resistance. The distance remains constant even as the interference strength increases, and the virus population rapidly decreases. Below this $\alpha$ threshold, however, a sufficiently high interference cost ($\kappa$) forces the virus to abandon the optimum and favor escape, resulting in a coevolutionary chase. This escape has direct consequences for viral control. In the "trapped" regime (high $\alpha$), increasing $\kappa$ effectively suppresses the viral population. In the "escape" regime (low $\alpha$), the virus population is less affected by $\kappa$ because it can simply evolve away from the interference with minimal loss of fitness.

Our model therefore suggests that the success of an evolutionary conflict depends critically on the fitness cost of escape mutations. The containment described by Rast et al. (2016) is effective because the pathways for virus escape, such as reduced capsid production, incur a large intrinsic fitness cost to the virus [27]. This scenario is represented by the high-$\alpha$ regime in our model, where the virus is successfully contained.

Our model demonstrates, however, that such containment may fail if the virus can find escape routes with a lower fitness penalty (the low-$\alpha$ regime). This scenario is exemplified by VSV, which evolved resistance to defective interfering particles through mutations in its polymerase complex. These mutations selectively reduced the affinity of the viral polymerase for interfering (DIP) templates while preserving its function on the viral genome, allowing escape with a minimal fitness cost [32]. This example highlights a fundamental asymmetry in the evolutionary arms race: the virus only needs to find one viable escape route within a vast, high-dimensional landscape of possibilities, whereas the TIP must be robust to all of them. Therefore, the long-term success of a TIP-based therapy likely depends on targeting a viral function so essential that any resistance mutation incurs a prohibitive fitness cost.

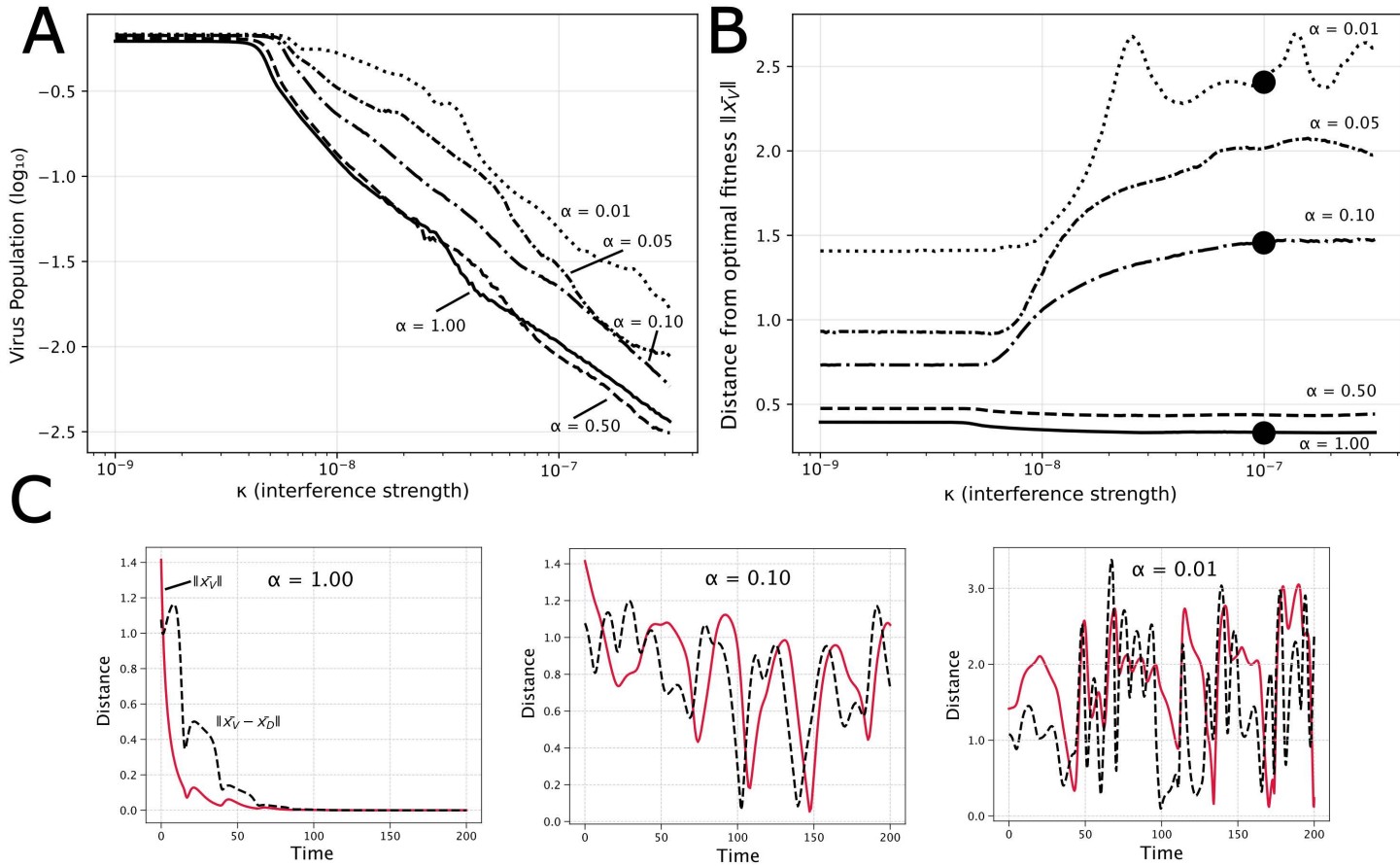

**Fig 6. Tradeoff between viral fitness and escape. (A)** Time-averaged virus population (normalized by $K$, log scale) over the final 20% of each simulation versus interference strength ($\kappa$), for fitness penalty values ($\alpha$) ranging from 0.01 (dotted) to 1 (solid). **(B)** Time-averaged distance of the virus centroid from the fitness optimum versus interference strength. High $\alpha$ values maintain viruses near the optimum regardless of $\kappa$ (flat lines), representing successful evolutionary containment. Low $\alpha$ values show increasing escape with interference strength. Black dots correspond to parameters in part **(C)**. **(C)** Time series of virus centroid distance from the fitness optimum $\|\bar{x}_V\|$ and distance between virus and DIP centroids $\|\bar{x}_V - \bar{x}_D\|$ for three values of $\alpha$ at $\kappa = 10^{-7}$. At high $\alpha$ (left), the virus is trapped near the optimum and both distances decay. At intermediate $\alpha$ (center), moderate oscillations in both distances occur. At low $\alpha$ (right), large, sustained oscillations indicate an ongoing coevolutionary chase.

## Discussion

We developed a mathematical model to investigate the coevolutionary dynamics between a virus and its defective interfering particles. By integrating phenotype-dependent interference, mutation, and intrinsic fitness landscapes, our framework demonstrates how a limited set of virus–DIP interactions can generate a wide spectrum of complex outcomes. The model successfully reproduces key phenomena observed in experimental systems, including oscillating population dynamics (stemming from the von Magnus effect) and sustained coevolutionary chases (Red Queen). The emergence of these behaviors from the model's core assumptions provides a powerful tool for exploring the evolutionary principles that govern viral resistance to interfering particle therapies.

### Parameter validation

While individual parameter values carry uncertainty, we emphasize two forms of validation. First, the model's emergent dynamics (oscillatory population cycles and transient resistance patterns) reproduce qualitative phenomena that have

been repeatedly observed in experimental systems [5–7,9,31,32]. More granular experimental evidence matches predictions as well. For instance, the shift from coexistence to chase dynamics as interference strength increases (Fig 4A) is supported by laboratory evolution studies. Pelz et al. tracked DIPs of influenza A in long-term culture and found that the variants that eventually dominated—and drove oscillatory population dynamics—were those with greater interfering efficacy [65]. This selection for highly potent DIPs in a dynamic context mirrors the "chase" regime of the model, which arises only when $\kappa$ is high enough to overcome coexistence constraints. Second, the qualitative regime structure (chase, coexistence, extinction) is robust across multiple independent tests of the model's structural assumptions. Specifically, extension to three trait dimensions produces similar behaviors (S1 Fig), alternative interference kernels preserve chase dynamics (S2 Fig), and the global sensitivity analysis explores outcomes across variation in all parameters simultaneously (Fig 4). These tests demonstrate that the regimes identified here are not artifacts of a single parameter combination or functional form. We also note that the model's biological conclusions are stated at the level of qualitative patterns rather than quantitative predictions, precisely because the abstract phenotype space precludes direct numerical calibration against any single experimental system.

## Model limitations

Our model's explanatory power is limited by its core structural assumptions. We defined viral interaction using a Gaussian interference kernel and implemented quadratic fitness terms ($\alpha, \beta$). While these are mathematical abstractions, the model's behavior is not wholly dependent on them; substituting alternative kernels (e.g., top-hat, exponential) produced qualitatively similar chase dynamics (S2 Fig), suggesting the results are robust.

Further simplifications relate to the model's scope. We represent the vast genetic and phenotypic landscape of a virus in only two dimensions. In reality, dimensionality is likely much higher when taking into account complex interference mechanisms, like the activation by DIPs of host immune defenses [12,66] or the creation of novel chimeric proteins and functions as a byproduct of DIP formation [28,67–72]. Here too, our core findings hold under limited testing: simulations in three trait dimensions showed similar behaviors to two dimensions, though computational costs prohibited a thorough analysis (S1 Fig). We use a deterministic framework that neglects the randomness of evolution, but assume that in large populations, the stochastic effects are less prominent. Our modeling of distance using simple Euclidean measures between population centroids omits information about the shape of the distributions. However, we find similar trends when using a more comprehensive distance measure (S3 Fig).

For instance, while our model qualitatively reproduces observed transient resistance patterns, there are quantitative differences from the experimental data (Fig 5). The modeled resistance changes span roughly 10- to 50-fold, whereas the experimental data of DePolo et al. (1987) show resistance peaks of $10^5$-fold. This quantitative gap is primarily a consequence of the two-dimensional phenotype space, which limits the maximum phenotypic separation the virus can sustain against the centering penalty ($\alpha$). Fold resistance is determined by the overlap between the virus and DIP distributions through the convolution integrals (Eq. 14), and this overlap decays as a Gaussian with separation in phenotype space. More phenotype dimensions could substantially amplify fold-resistance values by allowing greater separation, consistent with the model's extension to three dimensions (S1 Fig). Additionally, the model neglects the constant accumulation of mutations observed in VSV in regions like the N gene and 5' terminus, which prevent the virus from returning exactly to previous phenotypes [31]. This directional drift, representable as movement along additional phenotypic dimensions, would further increase fold-resistance magnitudes. These discrepancies are expected consequences of model simplifications, such as the two-dimensional phenotype space, which constrains evolutionary trajectories. Within our framework, the drift and orders-of-magnitude resistance could be represented as a "corkscrew" trajectory through higher-dimensional space, where cyclic pursuit in some dimensions is coupled to directional drift in others. Such an extension could be implemented by expanding the phenotype space or allowing the quadratic fitness optimum in the $\alpha$ term to drift slowly over time. While promising, these ideas lie beyond our present goal of isolating the core chase dynamics. The

model's value here is not in precise numerical replication, but in its ability to predict how the qualitative chase dynamic responds to changes in biological parameters.

Applying our modeling approach to a specific virus system will require translating its abstract phenotype space into one defined by concrete, measurable biological traits. For instance, the trait axes may represent features such as the viral replicase or capsid protein binding affinities to the virus and DIP genomes. Such a specialization would necessitate modifying the model's core assumptions. Mutation, for example, may no longer be isotropic, as certain traits may be inherently more sensitive to mutation than others. Furthermore, representing virus and DIP traits in separate phenotype spaces to capture virus- or DIP-specific parameters (e.g., DIP genome length) would demand a redesign of the interaction kernels. While these adaptations would enhance biological realism, they would do so at the cost of the model's generalizability. Therefore, the abstractions of the current model are best viewed as a foundational framework, designed to be adapted and specialized for future, virus and host specific investigations.

## Analytical tractability

The structural choices in the model—a quadratic fitness landscape and Gaussian interference kernel—are shared with phenotype-structured systems which have been analytically studied. Additionally, the lack of sensitivity to the mutation rate (Fig 4F) suggests that an asymptotic analysis may be feasible [36]. For single-population replicator-mutator equations with quadratic fitness, Gaussian distributions are preserved exactly under the dynamics, and the population mean, variance, and total size satisfy explicit ODEs [37,73]. Alfaro et al. (2025) used this same quadratic-Gaussian structure to prove the existence of traveling-pulse solutions in a related host-pathogen coevolution model [38]. In our coupled virus–DIP system, each species equation individually inherits this favorable structure; however, the coupling terms introduce complications that prevent a direct application of existing results. The interference interaction, a convolution of the Gaussian kernel with the opposing species' distribution, contributes a non-quadratic term to the effective fitness landscape, while the *de novo* generation term deposits virus-shaped density directly into the DIP equation, potentially producing bimodal distributions when the two populations are phenotypically separated. Developing analytical reductions that accommodate these features (e.g., a Taylor-series approximation of the interference term, a multi-component moment-closure ansatz, or an asymptotic study of mutation [36,74]) is a natural direction for future work that could yield closed-form conditions for the regime boundaries (chase, coexistence, extinction) identified numerically here. Such a reduction would open the door to optimal control approaches for therapeutic design, enabling principled optimization of TIP dosing and phenotype-targeting strategies that account for coevolutionary feedback.

Another natural generalization is to replace the diffusion operator $\mu\nabla^2$ with a nonlocal mutation kernel, representing mutation as an integral operator $\int J(\mathbf{x}-\mathbf{x}')v(\mathbf{x}',t)\,d^2\mathbf{x}' - v(\mathbf{x},t)$ rather than a Laplacian [75]. The diffusion approximation is formally valid only when mutation steps are small relative to the scale of the fitness landscape, as demonstrated by Champagnat et al. [76], who showed that the Laplacian arises as a specific scaling limit of such integral operators. However, some phenotypic changes relevant to virus–DIP systems, such as large deletions that generate DIPs or polymerase mutations that substantially alter template specificity, may correspond to large jumps in phenotype space rather than incremental diffusion. A nonlocal kernel could capture such discontinuous phenotypic transitions with fat-tailed mutation distributions where rare large-effect mutations play a disproportionate role.

## Experimental predictions

A clear path forward is to inform the parameters of our model with experimentally measurable quantities. Achieving this would require experiments designed to constrain individual parameters. Parameters like mutation rate ($\mu$) and decay rate ($\gamma$) could be directly manipulated in laboratory systems [77,78]. Others, such as the interference strength ($\kappa$), would require more sophisticated experimental designs, such as controlled coinfection experiments at varying multiplicities to quantify per-particle interference strength [43].

A compelling experimental test of our model's predictions would be to compare the coevolutionary dynamics of different virus–DIP systems. For example, bioreactor systems with higher outflow rates should, according to our model, exhibit more stationary evolutionary dynamics than those with lower rates (Fig 4E). Such comparative studies would provide a robust validation of the core principles identified in our theoretical framework.

Critically, our model predicts that for a given virus–DIP system, it should be possible to experimentally determine whether the interaction falls in the trapped or escape regime. In the trapped regime (high $\alpha$), viral populations should remain susceptible to contemporaneous DIPs across passages, with no directional drift in interference-relevant traits. In the escape regime (low $\alpha$), one would expect measurable phenotypic drift alongside transient resistance signatures in time-shift assays (Fig 5). Competition assays quantifying the fitness cost of known resistance mutations, combined with deep sequencing techniques, could place a specific virus–DIP pair on the $\alpha - \kappa$ landscape of Fig 6 and inform whether a TIP strategy requires additional design constraints to prevent evolutionary escape.

### Therapeutic implications

The sensitivity analysis reveals that successful TIP design may require navigating several trade-offs rather than simply maximizing any single parameter. The most robust therapeutic outcomes occur in parameter ranges with high interference and high costs of evolutionary escape (Fig 6). These lead to the ideal scenarios of: (1) complete virus extinction or (2) established stable coexistence with ongoing but manageable viral suppression. The intermediate chase regime, while evolutionarily interesting, may be less desirable therapeutically due to its oscillatory viral loads and potential for eventual viral escape. Critically, these results suggest that TIP failure is most likely to occur through low costs to escape (e.g., targeting non-conserved viral functions in high-mutation environments) or insufficient interference strength relative to viral fitness advantages.

In summary, this work presents a framework for understanding virus–DIP coevolution not as a simple arms race, but as a complex interplay of forces that can lead to diverse outcomes. The model's primary contribution is to translate abstract evolutionary pressures into distinct, predictable regimes. By identifying the conditions that lead to stable suppression, viral escape, or sustained coevolutionary chases, our model provides a conceptual foundation for generating testable hypotheses. Ultimately, this approach may help guide the design of therapeutic strategies that anticipate and steer viral evolution.

### Supporting information

**S1 Video. Video visualization of simulation in Figs 3A–3C.** Population density heatmaps with mean phenotype trajectories of virus (red) and DIP (blue) overlaid.
(MP4)

**S2 Video. Video visualization of simulation in Figs 3D–3G.** Population density heatmaps with mean phenotype trajectories of virus (red) and DIP (blue) overlaid.
(MP4)

**S1 Table. Convergence validation.** Confusion matrix comparing outcome classifications at $T=200$ versus $T=400$ for all 10,000 LHS simulations. Overall concordance: 9,286/10,000 (92.9%).
(PDF)

**S1 Fig. Higher dimensional modeling.** We extend the model to function in three dimensional space. We initialized the models with only a small perturbation in the 3rd dimension. From that, we saw that the perturbation expanded to the higher dimension, rather than any collapse or full escape, suggesting that the chase modeling can be stably extended to higher dimensionality.
(EPS)

**S2 Fig. Alternative interference kernels.** The main text model uses a Gaussian interference kernel ($e^{-\frac{\|\mathbf{x}\|^2}{2\sigma^2}}$), to capture the intuition that more similar DIPs will better be able to interfere with the virus. However, we relax this assumption and test several other kernels. These include exponential ($e^{-\frac{\|x\|}{\sigma}}$, slower decay) (d-f), rational ($\frac{1}{1+\|x\|}$, much slower decay) (g-i), and step function (1 if $\|x\| < \sigma$, else 0) (j-l). For the most part, these different kernels yield distinct but not disparate trajectories, suggesting that the model is robust to the choice of interference kernel.
(TIFF)

**S3 Fig. Earth mover's distance.** A simple Euclidean distance can be misleading, as it overlooks changes in the shape and spread of the population distributions. We therefore also compute the Earth Mover's Distance (EMD), which measures the total "work" required to transform the full virus distribution into the DIP distribution. The EMD provides a more comprehensive measure of their dissimilarity and also exhibits clear oscillations, validating that the chase dynamic involves the entire quasispecies, not just the movement of its mean.
(EPS)

**S1 Text. Nondimensionalization of the model equations.** Defines the dimensionless variables and parameter groups used to rescale the dimensional model (Eqs. 1–2), provides the full dimensionless system (Eqs. S1–S2), and summarizes all dimensionless parameters in a reference table. All parameter values reported in the main text are these dimensionless groups.
(PDF)

## Acknowledgments

J.Y. acknowledges institutional support from the University of Wisconsin–Madison, including the Wisconsin Institute for Discovery, the Office of the Vice Chancellor for Research, and the Department of Chemical and Biological Engineering. S.M. appreciates the support of Prof. JR Schmidt.

## Author contributions

**Conceptualization:** Shiv Muthupandiyan, John Yin.

**Data curation:** Shiv Muthupandiyan.

**Formal analysis:** Shiv Muthupandiyan.

**Funding acquisition:** John Yin.

**Investigation:** Shiv Muthupandiyan, John Yin.

**Methodology:** Shiv Muthupandiyan, John Yin.

**Project administration:** John Yin.

**Software:** Shiv Muthupandiyan.

**Supervision:** John Yin.

**Validation:** Shiv Muthupandiyan.

**Visualization:** Shiv Muthupandiyan.

**Writing – original draft:** Shiv Muthupandiyan, John Yin.

**Writing – review & editing:** Shiv Muthupandiyan, John Yin.

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
