## [Decision Letter · Decision Letter 0]

19 Feb 2026

PCOMPBIOL-D-25-02183

Coevolutionary dynamics of viruses and their defective interfering particles

PLOS Computational Biology

Dear Dr. Yin,

Thank you for submitting your manuscript to PLOS Computational Biology. After careful consideration, we feel that it has merit but does not fully meet PLOS Computational Biology's publication criteria as it currently stands. Therefore, we invite you to submit a revised version of the manuscript that addresses the points raised during the review process.

We look forward to receiving your revised manuscript.

Kind regards,

Tyler Cassidy

Academic Editor

PLOS Computational Biology

Zhaolei Zhang

Section Editor

PLOS Computational Biology

**Additional Editor Comments:**

Both reviewers appreciated this mathematically driven manuscript and raise important points to be addressed in revision. While mathematical models, such as the one presented in this work, can drive new biological knowledge and lead to novel insights into the underlying biological systems, I believe it is extremely important to justify the validity of the underlying mathematical model and clearly demonstrate that the conclusions being drawn reflect the biological reality, rather than arising from the mathematical model studied. Thus, this manuscript may require significant revisions to further establish the robustness of the model-derived conclusions.

In particular, it is not clear how the model parameters were chosen, nor how these could be inferred from the experimental system described in Figure 1. As many of the results in this manuscript are obtained via model simulation, it is extremely important to justify where these parameters are coming from, and why they are appropriate, though I appreciate that the authors have performed a 1 by 1 sensitivity analysis. This is particularly relevant to ensure that the conclusions drawn in this manuscript drive new biological understanding, rather than mathematical artifacts of the model framework chosen. Consequently, it will be important to fully address Reviewer 2's 2nd point. Similarly, it is important to further justify the modelling decisions made when developing the phenotypically structured model. While the current modelling choices may allow for an analytical investigation of the model (as reviewer 1 notes), as the model is not validated against experimental data, it is increasingly important to either strongly justify these modelling choices or show that the biological conclusions drawn from this model are robust to alternative model formulations.

Finally, I note PLOS Computational Biology's policy on the use of artificial intelligence (https://journals.plos.org/plosone/s/ethical-publishing-practice#loc-artificial-intelligence-tools-and-technologies) :

Contributions by artificial intelligence (AI) tools and technologies to a study or to an article’s contents must be clearly reported in a dedicated section of the Methods, or in the Acknowledgements section for article types lacking a Methods section. This section should include the name(s) of any tools used, a description of how the authors used the tool(s) and evaluated the validity of the tool’s outputs, and a clear statement of which aspects of the study, article contents, data, or supporting files were affected/generated by AI tool usage.

**Journal Requirements:**

At this stage, the following Authors/Authors require contributions: Shiv Muthupandiyan, and John Yin. Please ensure that the full contributions of each author are acknowledged in the "Add/Edit/Remove Authors" section of our submission form.

3) Some material included in your submission may be copyrighted. According to PLOSu2019s copyright policy, authors who use figures or other material (e.g., graphics, clipart, maps) from another author or copyright holder must demonstrate or obtain permission to publish this material under the Creative Commons Attribution 4.0 International (CC BY 4.0) License used by PLOS journals. Please closely review the details of PLOSu2019s copyright requirements here: PLOS Licenses and Copyright. If you need to request permissions from a copyright holder, you may use PLOS's Copyright Content Permission form.

Potential Copyright Issues:

i) Figure 1. Please confirm whether you drew the images / clip-art within the figure panels by hand. If you did not draw the images, please provide (a) a link to the source of the images or icons and their license / terms of use; or (b) written permission from the copyright holder to publish the images or icons under our CC BY 4.0 license. Alternatively, you may replace the images with open source alternatives. See these open source resources you may use to replace images / clip-art:

4) Please amend your detailed Financial Disclosure statement. This is published with the article. It must therefore be completed in full sentences and contain the exact wording you wish to be published.

2) If any authors received a salary from any of your funders, please state which authors and which funders..

**Reviewers' comments:**

Reviewer's Responses to Questions

**Comments to the Authors:**

Reviewer #1: See attachment

Reviewer #2: In this interesting paper, the authors consider a model of defective interfering particles (DIPs) that interfere with the spread of a viral infection. I had not been aware of these prior to reading this paper. They develop a model of how the viruses evolve in the presence of DIPs and examine and discuss the resultant dynamics. From a dynamical systems perspective the model has similarities to predator-prey models, as the authors point out.

This is a well written paper which explains the model and results clearly. The model development seems logical, with the different modelling components explained carefully and well within the Methods section. The results seem plausible and the paper has a clear and convincing narrative accompanied by very informative figures. Figures 3 and 4, and the accompanying videos in the SI, nicely illustrate the dynamics effects that can occur, including the chase scenario that they describe. Figure 5 shows the effect of varying parameters; the outcome of the low washout rate in particular leading to greater extinction probability was initially surprising to me until reading the explanation.

I have a few points where I would like the authors to clarify and potentially amend the paper but otherwise I think that this is a very good piece of work that warrants publication in PLOS Computational Biology.

1) Equations (12,13) are almost identical to equations (10,11). The only difference is that the parameter rV is replaced by 1, a scaling that could easily have been done at the start by suitably defining the time units. Thus it seems that there is some unnecessary work towards the end of the Methodology section, which hopefully the authors can streamline.

2) At the bottom of page 7 some specific numerical values are selected. Why these numbers in particular/ how robust are results if these are changed? We see later in the paper what happens when single parameter values are varied in turn, but what if a completely new set of parameters was used?

3) In Figure 6 and its associated caption, the terms "fold reduction" and "fold resistance" are used. These are mentioned nowhere else, however, and not explained. What do they mean and are these different things? What is their relevance?

4)In Figure 7 A, can the authors explain the meaning of the line crossings, and why they occur?

**Have the authors made all data and (if applicable) computational code underlying the findings in their manuscript fully available?**

Reviewer #1: Yes

Reviewer #2: Yes

PLOS authors have the option to publish the peer review history of their article (what does this mean?). If published, this will include your full peer review and any attached files.

**Do you want your identity to be public for this peer review?** For information about this choice, including consent withdrawal, please see our Privacy Policy.

Reviewer #1: **Yes:** Chiara Villa

Reviewer #2: No

**Figure resubmission:**
---

## [Decision Letter · Decision Letter 1]

6 May 2026

Dear Dr. Yin,

We are pleased to inform you that your manuscript 'Coevolutionary dynamics of viruses and their defective interfering particles' has been provisionally accepted for publication in PLOS Computational Biology.

Best regards,

Tyler Cassidy

Academic Editor

PLOS Computational Biology

Zhaolei Zhang

Section Editor

PLOS Computational Biology

Reviewer's Responses to Questions

**Comments to the Authors:**

Reviewer #2: I liked the first version of this paper and the authors have addressed the small number of points from my previous review well. The only thing I saw in the new version for them to look at is that in the caption of Figure 1, the left panel is mentioned, but not the right one.

**Have the authors made all data and (if applicable) computational code underlying the findings in their manuscript fully available?**

Reviewer #2: None

PLOS authors have the option to publish the peer review history of their article (what does this mean?). If published, this will include your full peer review and any attached files.

Reviewer #2: No

---

## [Editor Report · Acceptance letter]

PCOMPBIOL-D-25-02183R1

Coevolutionary dynamics of viruses and their defective interfering particles

Dear Dr Yin,

I am pleased to inform you that your manuscript has been formally accepted for publication in PLOS Computational Biology. Your manuscript is now with our production department and you will be notified of the publication date in due course.

With kind regards,

Anita Estes
